# Are Spaniards Happier When the Bars Are Open? Using Life Satisfaction to Evaluate COVID-19 Non-Pharmaceutical Interventions (NPIs)

**DOI:** 10.3390/ijerph181910056

**Published:** 2021-09-24

**Authors:** Pablo de Pedraza, María Rosalía Vicente

**Affiliations:** 1European Commission, Joint Research Centre (JRC), 21027 Ispra, Italy; 2Applied Economics, University of Oviedo, 33006 Oviedo, Spain; mrosalia@uniovi.es

**Keywords:** COVID-19, NPIs, life satisfaction, mental health

## Abstract

The COVID-19 pandemic has challenged governments worldwide with the design of appropriate policies that maximize health outcomes while minimizing economic and mental health consequences. This paper explores sources of individuals’ life satisfaction during the COVID-19 pandemic, paying special attention to the effects of non-pharmaceutical interventions (NPIs). We studied the specific case of Spanish regions and focused on bar and restaurant closures using data from a continuous voluntary web survey that we merged with information about region-specific policies that identified when and where bars and restaurants were closed. We estimated an endogenous binary-treatment-regression model and found that closing bars and restaurants had a significant negative impact on happiness. The results were statistically significant after controlling for the pandemic context, health, income, work, and other personal characteristics and circumstances. We interpreted the results in terms of the positive effect of socialization, individuals’ feelings of freedom, and the comparative nature of life satisfaction.

## 1. Introduction

The rising number of COVID-19 cases and deaths along with the prolonged lockdowns, the substantial restrictions on public life, and the subsequent economic downturn have affected personal well-being and mental health [1,2,3,4]. Governments worldwide are confronted with an unprecedented crisis with the COVID-19 pandemic. They face the challenge of designing policies that slow down the growth rate of infections with the minimum pernicious economic, social, and mental health effects [5]. This paper studied the effect of COVID-19 non-pharmaceutical interventions (NPIs) on individuals’ subjective well-being using Spanish regions as a case study and exploring the relationship between bars and restaurant closures and life satisfaction.

Our approach, using life satisfaction as the outcome variable to evaluate COVID-19 policies, assumed that life satisfaction is a legitimate goal for governments during the pandemic [6]. Several reasons are argued for its relevance.

First, life satisfaction scores reflect subjective and objective circumstances, personal characteristics, and contextual circumstances at regional and national levels [7,8,9,10]. Major events in work and family life, such as the loss of loved ones, divorce, or job loss, affect satisfaction levels [11]. Therefore, life satisfaction is likely to be affected by the pandemic itself and the containment policy measures such as prolonged lockdowns, restrictions on public life, the subsequent economic downturn, and specific personal circumstances. NPIs may have had a positive effect, as they reduced the initial uncertainty [12], but, in general, restrictions have caused detrimental effects on mental health [13].

Second, while the first goal of the restriction measures has been to contain the spread of the virus, their socio-economic side effects (those on life satisfaction included) are also important. There is an open discussion on the most effective policies to contain the spread of the virus. Specifically, there is a debate on whether stricter lockdown policies are associated with lower mortality [14]. Some studies report that stricter measures, such as lockdowns, are associated with lower transmissions and fewer deaths [15,16]. Other studies report quantification problems and that less disruptive and costly NPIs can be as effective as more intrusive and drastic ones [17,18,19,20,21,22,23]. We neither aim to make a detailed literature review of NPIs’ effectiveness on the spread of the disease nor to engage in the above discussion. However, given the lack of scientific consensus on NPIs’ efficiency on health outcomes, it might be useful to explore their social and economic consequences and side effects. Under the uncertainty that two-policy measures are equally efficient against the spread of the virus and mortality, policy makers may choose the one with a smaller impact on life dissatisfaction. In this sense, our focus on well-being (i.e., life satisfaction) complements the evaluation of NPIs in terms of health outcomes.

Third, economic hardship, context, and business performance also matters to mental health and life satisfaction [8,10,24,25]. Several economists have advocated for the design of targeted lockdowns in order to contain the virus while facilitating some degree of economic activity [5] that minimizes both COVID-19’s effects in terms of the number of cases and deaths and the economic and social impact on job destruction, unemployment, levels of depression, anxiety, and dissatisfaction.

Fourth, although the mechanisms by which mental health interacts with the immune system are not yet clear, literature has shown that emotional states contribute to the resistance or vulnerability to illness, unhappiness being a potential contributor to disease risk [26,27,28]. If happiness is protective, interventions targeting to minimize the negative impact on happiness may have a favorable impact on health and the evolution of the pandemic. Hence, life satisfaction may be an important piece of the puzzle to evaluate NPIs. 

During this pandemic, countries have implemented different strategies with various timings [25,29], and there is also within-country variation in the policies applied. For example, in the US, each state decided how, when, and what policies should be implemented. This opened the possibility to evaluate policy measures reducing cross-country unobservable differences and other difficulties of cross-country comparisons. Recent research shows that specific regional and state-level contexts influence physical and mental health outcomes related to the COVID-19 pandemic [1,30]. Donnelly and Farina (2021) [1], for example, report state-level variations in the effects of COVID-related income shocks on mental health. Specifically, people who lived in states with supportive social policies coped better with income shocks than those who did not.

Our case of analysis, Spain, is one of the European countries worst hit by the COVID-19 pandemic. During the first wave (March–June 2020), the responsibility of anti-contagion measures relied on the central government. After the summer of 2020, regional authorities were responsible for setting their own policies [31]. Therefore, a range of measures were implemented across Spain. Regions with similar levels of contagion opted for different NPIs.

Among all the policies, there were several reasons that motivated our decision to specifically study bar and restaurant closures in Spain. The Spanish economy is heavily oriented to tourism and hostelry services, which was the sector with higher employment losses [32]. Bars and restaurants play an important role in Spanish culture and social capital. Social capital is related to health [33]. Bars may also play a role in tackling several pandemic side effects such as loneliness [34]. It is illustrative that the closure of bars and restaurants was important to the political debate. Madrid and other regions kept bars and restaurants open throughout the second wave, while Catalonia, and other regions, with similar levels of contagion opted for their closure during specific periods [35,36]. 

Based on all the above, we hypothesized that, after accounting for the pandemic context, income, work, and family personal circumstances, bars and restaurant closures negatively affected individuals’ levels of life satisfaction. We also explore whether the effect is conditional on personal specific circumstances. For some people, such as those with low self-perceived health, the effect might have been positive because restrictions lower uncertainty about the evolution of the contagion. For others, such as those losing their job, closures might have had a negative effect because of the negative economic consequences.

For that purpose, we used data collected by means of a web survey from September to December 2020. Endogenous treatment-regression models were estimated, life satisfaction being the outcome variable. The policy applied defines the treatment. To study the differences in the policy effects among respondents with different health and labor status, we used interaction terms. We found that bar and restaurants closures displayed a significant negative effect on life satisfaction and that the effect was not conditional on personal characteristics such as health and labor status. This paper contributes to understanding the channels through which the evolution of COVID-19 and the political decisions taken have affected citizens. It shows that policies allowing for a certain degree of economic activity and socialization may counteract the negative effects that the pandemic is causing on life dissatisfaction. The remainder of the paper is structured as follows: Section 2 explains the data, variables, and estimation methodology; Section 3 explores the results; Section 4 presents the discussion; Section 5 concludes the paper.

## 2. Materials and Methods

### 2.1. Data

This paper used microdata from the survey Living and Working in Coronavirus Times (LWCV) in Spain from 1 September through 17 December 2020. This survey was the result of an international project launched to collect data on people’s living and working conditions during the COVID-19 pandemic [37]. The 10 min online survey included a range of questions to measure individual and family interpersonal coping with COVID-19 and its impact on individual work situations and family income. Survey participation was promoted via social media, press releases, snowballing, messages in widely distributed newsletters, and websites of partners. Data collection started on 23 March 2020 and will continue as long as the pandemic lasts.

During the period of analysis, a total of 4010 respondents completed the Spanish questionnaire. The sample included 50% women; the average age of the respondents was 46.7 years with 50% of the sample being older than 50 years. 46% of respondents had a low level of education, almost 20% a middle level of education, and 34% had a high level of education. Approximately 40% lived with one or more children, and 61% lived with a partner. Table 1 shows the distribution of respondents per region.

Though data are available from an early period (from March 2020 on), we focused our analyses on the second COVID-19 wave. During the first wave, restrictions were implemented by the central government and, thus, were homogeneous across regions. During the period of study, regions under similar pandemic circumstances were implementing different containment measures which allowed, to a certain extent, for the isolation of the effect of the measure from the pandemic itself.

It is important to consider that the LWCV survey is a voluntary, continuous web-survey. The main advantages of web surveys are that they can be implemented quickly and then allow for the study of rapidly changing situations such as the current pandemic. Indeed, several previous studies on the impact of COVID-19 on health, work, and personal and family situations have used data from the LWCV or other web surveys [3,4,12,38,39,40,41]. The main drawback of voluntary web surveys is that they might suffer from some self-selection biases. In this paper, potential biases were studied by comparing the main socio-demographic features of our sample with the data from the Spanish Labor Force Survey (Encuesta de Población Activa, EPA) for the third-quarter of 2020 [42]. Following a conventional statistical procedure [43], we calculated weights using data from the Spanish Labor Force Survey to balance our sample and correct it for potential selection biases in terms of age, gender, and region. 

To explore the effects of regional-specific restriction measures, we merged the LWCV survey data with several other data sources at regional level, first, being data on the daily number of COVID-19 cases [44]. The National Health Institute Carlos III is a public institution in charge of collecting COVID-19-related statistics in Spain. Second, a specific database was built by gathering information on the measures adopted during the pandemic at the regional level. This information was collected from the Official Journals of each region in which, by law, any policy measure should be published before implementation. The information published included the context and reasons for the policy, as well as the time and territory of application, and any exceptions that might apply. Over the period of analysis, 12% of the respondents were living in a region where bars and restaurants were closed at the moment of the interview due to the restrictions imposed by the regional government. 

### 2.2. Variables

Table 2 lists the variables used in the analysis. The variable of interest was individuals’ level of life satisfaction (SATLIFE). The LWCV survey requested respondents to report their overall level of life satisfaction on a scale from 1 (minimum level of life satisfaction) to 10 (maximum level of life satisfaction).

The NPI analyzed bars and restaurants closures through is a dummy variable (BARS_CLOSE) that equaled 1 if the region where the respondent lived was under this particular restriction at the time when the respondent completed the survey. It is important to take into account that the closedown of restaurants/bars is a homogenous measure across regions, in the sense that there were only two options, bars were closed or not. Keeping the bars open implied the provision of outdoor space so as to comply with capacity restrictions, normally set between 50% and 75%, and other recommendations, such as closure at 23:00, or limitations on the amount of people sharing a table to a certain number [31]. In order to have additional outdoor space, bar owners could use public parking spaces that, because of the sharp reduction in traffic and commuting, were no longer needed. This slightly changed the landscape of cities under this policy enabling socialization. Regions like Catalonia, Valencia, Basque Country, Asturias, and Navarra imposed bar closures. Regions like Madrid, Galicia, Cantabria, Aragón, and Extremadura shifted responsibility to individuals. The media coverage made it possible for citizens to be aware of the different situations of their counterparts. People living in Barcelona, Valencia, and the Basque Country could see how in Madrid and other regions people were able to spend time with their friends and family and socialize while keeping to certain rules, while their own region’s bars were closed and their *back to normal* was postponed repeatedly. As previously indicated, 12% of respondents completed the questionnaire in a moment when bars and restaurants were closed in their region while open in others. Both treatment and control groups were distributed similarly in terms of gender, age, education, self-perceived health, and other life satisfaction determinants.

The daily number of positive cases per region (CASES) measures the levels of COVID-19 contagion in each region. We used this variable to model the decision to close bars and restaurants.

Along with the potential effects of restriction measures on life satisfaction, we controlled for socio-demographic features: age, gender, educational attainment, living with a partner, children, employment situation, health status, and healthy lifestyle. In addition, we considered individuals’ personal, family, work, health, and income circumstances. Two binary indicators taken into account were whether the respondent was currently staying home due to the COVID-19 restrictions and whether the respondent felt lonely. Another two dummy variables summarized the respondents’ and family health situations in relation with to COVID-19: whether he/she has recovered from the disease and whether any family member had been diagnosed. Finally, two other dummy variables assessed whether the respondent had lost a job due to the crisis and whether he/she thinks their future income will be lower. 

### 2.3. Empirical Approach

We aimed to hypothesize that bar and restaurant closures negatively affected individuals’ levels of life satisfaction. First, because citizens who would be able to slightly return back to normal, although still under the aforementioned limitations, would probably experience a sense of social and psychological freedom which is related to higher levels of happiness [45]. Second, we know that happiness is relative and comparison with others determines levels of subjective well-being [46]. Citizens in regions with stricter policies probably compared themselves with those in regions where the amount of social life relied on individual freedom. The fact that pandemic numbers reported in the media did not support stricter policies probably reinforced feelings of comparative lack of liberty. 

We also explored whether the effect of bar and restaurant closures were conditional on personal specific circumstances. While keeping bars open can have a positive effect for some people with strong self-perceived health, the measure may generate anxiety among those with lower self-perceived health. Similarly, for those that have lost their job and are willing to see the economy functioning, closures might have a stronger negative effect because of its negative economic consequences. 

To capture the effects of NPIs on individuals’ life satisfaction, we followed an estimation strategy able to address two issues. The first issue was that the implementation of NPIs cannot be considered exogenous. Specifically, these policies were used to contain the spread of the virus when the figures for the contagion were rising and the pressure on health services was increasing. This first feature creates a setting similar to that addressed by the endogenous treatment-regression models with a binary-treatment variable (=1, under the NPI “bars and restaurants closed”; =0, otherwise). This type of model considers a linear potential-outcome regression which allows for some correlation structure between the unobservables that affect the treatment and those affecting the potential outcomes. The second issue is that the treatment variable (i.e., “bars and restaurants closed”) is at the regional level and not at the individual level. In other words, the treatment of interest is a feature of the region wherein the individual lives; it varied through time but did not vary among the people living in the same region at the specific moment when the policy was being implemented. A similar problem arises in Health Economics literature when the effect of a binary endogenous treatment is estimated at the provider level on a patient-level outcome [47]. Additionally, regions which choose to implement this policy might be different in some unobservable characteristics from those territories that do not (e.g., resources, regional government’s attitudes towards the pandemic). To solve this issue, we needed an instrument variable that was correlated with treatment but not with individual outcome, i.e., the instrument should have no direct effect on the outcome other than through the treatment. In our case, we used the number of daily cases per region as an instrument that would allow us to model the probability that a region decides to close its bars and restaurants. To check the appropriateness of this instrumental variable, we regressed SATLIFE on CASES. Estimates showed that CASES was not statistically significant (see Appendix A, Table A1). Therefore, our method fulfilled the basic assumption for instruments to be correlated with the treatment but not with the individual outcome. 

Taking all this into account, our empirical model is the following: 

Outcome equation:SATLIFE_ih_ = X_i_β + δT_h_ + ε_ih_(1)

Treatment equation:T_h_ = 1, if Z_h_γ + u_h_ > 0 (0, otherwise)(2)
where SATLIFE is the outcome of interest, i.e., the level of self-reported life satisfaction on a scale of 1–10 by an individual i, living in a region h; X are the variables that shape individuals’ life satisfaction; T is the treatment applied in the region h, wherein the individual lives; Z is the instrumental variable that determines the treatment assignment, i.e., whether the region implements some restrictive measure j, or not; ε_ih_ and u_h_ are the error terms of the outcome and treatment equations, respectively. They are bivariate normal with mean zero and covariance matrix $\left[\begin{array}{cc} \sigma^{2} & \rho \sigma \\ \rho \sigma & 1 \end{array}\right]$. The covariates of the outcome and treatment equations must be orthogonal to these error terms. We estimated the equations by the maximum likelihood. 

The use of endogenous treatment effects was justified because standard treatment effects could not be applied, since the phenomena we were studying did not meet two standard treatment effects estimator’s assumptions. On the one hand, the independent and identically distributed sampling assumption requires that the outcome and treatment status of each individual are unrelated to all other individuals. To a certain extent, our approach did not meet this assumption, since the treatment status, although different for individuals living in different regions, is the same for all individuals living in the same region. On the other hand, the conditional-independence assumption implies that potential outcomes are independent of the treatment assignment once we control for all observable variables. Neither is this assumption fulfilled because of the endogenous nature of the treatment assignment.

## 3. Results

Table 3 shows the results of the estimation of the outcome and treatment equations of our endogenous binary-treatment-regression model. The last rows of the table report the ancillary parameters of the estimation and the likelihood-ratio test for the null hypothesis of no correlation between the treatment-assignment errors and the outcome errors. The *p*-value of this test clearly suggests rejecting this null hypothesis, a result which supports the chosen empirical strategy.

Regarding treatment, the level of COVID-19 spread, measured by the number of daily cases, clearly determines the implementation of the restrictive measure. The variable CASES is statistically significant at the 1% level with a positive sign for the policy measure considered, the closing of restaurants and bars. Hence, the higher the number of daily cases in a region, the more likely that the regional government decided to close bars and restaurants. Alternative specifications were considered including square terms of CASES and regional and time dummies; however, those specifications posed convergence problems.

Regarding the effects of the policy of closing bars and restaurants, estimates showed that it was statistically significant with a negative sign. Then, bars and restaurants closures were related with a lower degree of life satisfaction of individuals living in the regions where this policy was applied. Table 4 shows that none of the estimated interaction terms between this policy and income and work personal situations derived from the pandemic, namely, job loss and income reduction, were significant. Estimated interaction terms between this policy and health and loneliness were also not significant. The negative effect of bar closures on life satisfaction was independent from personal circumstances. There was no specific group that was more or less affected than others.

Regarding individuals’ socio-economic control variables, the results are in line with what is generally assumed in life satisfaction literature before and during the pandemic. Estimates show that the levels of life satisfaction of Spaniards during the period of analysis (September–December 2020) was positively (and statistical significantly) related to having higher education, living with a partner, getting enough daily exercise, and having a job. The level of life satisfaction was negatively (and statistical significantly) related to self-reported health status: people with poorer health status tended to report lower levels of life satisfaction compared to those with better health. There was also some statistically significant evidence of the negative role of loneliness, job loss, and expectations of income loss. However, other COVID-19-related issues, such as being confined or having suffered the disease themselves or a family or friend, were not statistically significant.

## 4. Discussion

The goal of the paper was to evaluate the relationships between COVID-19 NPIs and life satisfaction. We evaluated bar and restaurant closures in Spain. The corresponding variable identifies citizens that, by the time of response, were living in a region where, although subject to certain capacity restrictions, bars and restaurants were open and offering some outside space for social interaction. The main finding of this paper is the negative and statistically significant association found between the closures of bars and restaurants and people’s life satisfaction. People living in regions where policies allowed for some certain degree of outdoor social interaction were significantly happier than those where the bars and restaurants had been closed. Keeping bars and restaurants open appeared to significantly counteract the negative effects on mental health and well-being that the pandemic and its policies are causing. The literature has shown that meeting with friends during leisure time has a positive effect on life satisfaction, while internet and tv consumption are negatively correlated with life satisfaction [48]. Keeping bars and restaurants open increases the possibility of meeting friends and socializing, which explains our results.

We did not find any significant association of the interaction terms. According to our results, keeping bars and restaurants open had a positive effect on life satisfaction regardless of personal characteristics and circumstances. The effect was not conditional on job and income loss, health status, or loneliness.

We believe that these results are related to the well-known positive effect of individual’s feelings of freedom on life satisfaction. We base this interpretation on previous literature that has shown that all kinds of freedom are related to happiness. Freedom and feelings of freedom are able to explain differences in life satisfaction across countries [45]. The possibility of making their own decisions, make citizens happier. In regions where bars and restaurants were open, those with poorer health status could decide to stay at home; keeping bars open was not a source of dissatisfaction for them. Those feeling heathier or lonely could socialize, although probably not as much as they would like. Both types of people could decide to take into account their personal characteristics and circumstances, which probably made them happier than their closed down counterparts. 

The comparative nature of life satisfaction also offers an explanation. The literature has shown that, with respect to income, happiness is comparative; it depends on the gap between an individual’s income and the benchmark to which him/her compares with [46]. People living in regions implementing stricter measures could witness the freedom of others. They could also see how their resignation from social life did not have clear compensation in terms of the number of cases and deaths. Therefore, happiness may be also comparative with respect to freedom. 

The rest of the results we obtained were, in general, consistent with previous evidence. First, regarding the importance of work, income, and personal circumstances on people’s well-being during the pandemic, job loss was the most important source of distress during this pandemic (note that its estimated coefficient was the highest in absolute value in the regression). Specifically, job loss was associated with a reduction in the level of life satisfaction by 0.8 points. Losing one’s job triggers a series of factors that negatively affect people’s well-being such as social and financial withdrawal and self-esteem problems, among others [1,4,25]. Escudero-Castillo et al. (2021) [38] found that compared to employed people, those unemployed or on furlough due to the pandemic tended to exhibit a higher risk of mental health problems. In addition, future income reduction due to the fact of COVID-19 negatively shaped people’s level of life satisfaction. Conversely, to be employed during the pandemic was positively associated with life satisfaction. 

Second, the results confirmed the well-documented link between well-being and health status [49]. Indeed, health status showed, after job loss, the strongest association with life satisfaction (the estimated coefficients for the former and the latter were −0.83 and −0.72, respectively). People reporting a bad health status tended also to report lower levels of life satisfaction. 

Third, along with job loss, income reduction, and health, we found that the feeling of loneliness was another major element lowering individuals’ life satisfaction. Though our data were posterior to the general lockdown of the population, people have been suffering from loneliness throughout the entire pandemic [50,51]. 

Fourth, daily exercise was positively associated with life satisfaction during the pandemic as previously shown in the literature [3]. 

Fifth, our results were only partially consistent with the U-shaped relationship between age and life satisfaction [52]. The estimated coefficients for age and its square term were negative and positive as corresponds to a U-shaped relationship. However, in our calculations, they were not significant. This was quite an interesting finding, because the elderly population is more vulnerable during the pandemic.

Although previous analyses showed that having family or friends diagnosed with COVID-19 was negatively associated with life satisfaction, our results showed that neither having suffered oneself from the disease nor having relatives/friends who had was associated with the level of life satisfaction. Such a discrepancy in the results could be due to the different time periods analyzed. During the first wave of the pandemic, the spread of the disease skyrocketed day by day [3], but our analysis considered the second wave, when both people and health services knew more about the disease and the figures of contagion and deaths never reached the levels of the first wave. Accordingly, having some family or friends diagnosed with COVID-19 during the first wave might have significantly affected someone’s level of life satisfaction due to the complex circumstances involved, while in the second wave the importance of this fact might have been lower. 

Some limitations should be noted. Though our study was carefully designed, and the variables and the empirical strategy were carefully chosen, some unobserved confounders might still exist that influenced our results. In this sense, it was important to take into account that bars and restaurants were closed in specific regions. Though the culture towards bars and restaurants is fairly similar across the Spanish territory, there might still be some regional cultural, political, and environmental differences in life satisfaction rather than the closure of bars. In what refers to NPIs, the closure of bars and restaurants was a measure added to a series of all other types of restrictions previously implemented (mobility restrictions, capacity limits in public spaces, etc.) and still active when bars and restaurants were closed. In addition, the regional governments that kept bars and restaurants open might tend to support individual freedoms through different policies. Therefore, it is possible that differences in life satisfaction might be due to the variations in the complex regulatory environments across regions, rather than the exclusive effect of keeping bars and restaurants open/closed. Moreover, we focused on a specific time period: the second wave of the pandemic. Over the subsequent waves, the importance of restrictions on life satisfaction might have changed.

Future research should then explore the several waves of the pandemic and analyze the impact of other policies on people’s well-being and whether their impact on happiness change over time. 

## 5. Conclusions

The COVID-19 NPIs aim to maximize health outcomes while minimizing their economic, social, and mental health consequences. Whether stricter lockdown policies are associated with lower transmission and mortality is an open discussion. Uncertainty on whether stricter policies are more efficient in reducing mortality makes side effects in the economy and mental health relevant for policy making. In addition, if happiness is protective, interventions targeting to minimize the negative impact on mental health and happiness may have a favorable impact on health and the evolution of the pandemic. When there is uncertainty about the relative efficiency of two policies, policy makers may choose the one with a smaller negative impact on life dissatisfaction. Assuming that the primary goal is to contain the virus, life satisfaction is a legitimate goal for governments and evaluating NPIs in terms of their impact on happiness is useful.

Against that background, this paper explored sources of individuals’ life satisfaction during the COVID-19 pandemic, paying special attention to the effects of NPIs. It studied the specific case of Spanish regions and focused on bar and restaurant closures using data from a continuous voluntary web survey merged with additional data sources regarding specific regional policies. We estimated endogenous binary-treatment-regression models.

We found that respondents living in a region where bars and restaurants were closed were significantly unhappier. The results were statistically significant after controlling for income, work, and personal shocks derived from the pandemic. During the second wave of the pandemic, work, income, feelings of loneliness, and the implemented policies shaped life satisfaction. Overall, our results highlight the importance of socialization and that policies relying on citizens’ free decisions rather than centralized stricter measures might be able to ameliorate the psychological damage of the pandemic. 

## Figures and Tables

**Table 1 ijerph-18-10056-t001:** Distribution of respondents by region.

Regions	Number of Respondents	% Percent
Andalucia	670	16.71
Aragón	149	3.72
Asturias	117	2.92
Baleares	171	4.26
Canarias	246	6.13
Cantabria	67	1.67
Castilla La Mancha	144	3.59
Castilla-León	249	6.21
Catalonia	589	14.69
Ceuta-Melilla	11	0.27
Comunidad Valenciana	351	8.75
Extremadura	120	2.99
Galicia	154	3.84
La Rioja	40	1
Madrid	608	15.16
Murcia	134	3.34
Navarra	76	1.9
Vasque Country	114	2.84
Total	4,010	100

**Table 2 ijerph-18-10056-t002:** Definition of variables.

Variable	Definition
SATLIFE	Respondents’ self-assessed level of life satisfaction on a scale of 1–10 (1 = Very dissatisfied, 10 = Very satisfied)
FEMALE	=1, if female
AGE	Age, number of years
TERTIARY_EDUCATION	=1, if tertiary education
HEALTH_STATUS	Respondents’ self-assessed health status on a scale of 1–5 (1 = Very good, 5 = Very bad)
L_PARTNER	=1, if respondent lived with a partner
L_CHILDREN	=1, if respondent lived with one or more children
CONFINED	=1, if respondent was staying home due to the corona restrictions at the time of the survey
EXERCISE	Respondents’ level of agreement with the following statement: “I get enough daily exercise” on a scale of 1–5 (1 = Completely disagree, 5 = Completely agree)
LONELY	Respondents’ level of agreement with the following statement: “I feel lonely in times of the corona crisis” on a scale of 1–5 (1 = Completely disagree, 5 = Completely agree)
JOB	=1, if respondent had a paid job
JOB_LOSS	=1, if respondent reported that he/she had lost a job due to the fact of COVID-19
FUTURE_LOWERINCOME	=1, if respondent expected that in the next month, he/she would receive less income
COVID19	=1, if respondent had recovered from the disease
COVID19_FAMILY/FRIENDS	=1, if respondents’ families or friends had been diagnosed with the disease
CASES	Daily number of COVID-19 cases per region
BARS_CLOSE	=1, if the region where the respondent lives had bars and restaurants closed at the time of the survey

**Table 3 ijerph-18-10056-t003:** Endogenous treatment-regression model of life satisfaction on bar and restaurant closures. Estimated coefficients.

Outcome Equation	SATLIFE
FEMALE	0.0864
AGE	−0.0335
AGE^2^	0.0005 *
TERTIARY_EDUCATION	0.2679 ***
HEALTH_STATUS	−0.7286 ***
L_PARTNER	0.4122 ***
L_CHILDREN	0.0456
CONFINED	0.0270
EXERCISE	0.1769 ***
LONELY	−0.5004 ***
JOB	0.3723 ***
JOB_LOSS	−0.8881 ***
FUTURE_LOWERINCOME	−0.5400 ***
COVID19	0.2884
COVID19_FAMILY/FRIENDS	0.0660
BARS_CLOSE	−2.6448 ***
Constant	9.2533 ***
**Treatment Equation**	**BARS_CLOSE**
CASES	0.0002 ***
Constant	−1.3598 ***
/athrho	0.7952
/lnsigma	0.7296
rho	0.6614
sigma	2.0742
lambda	1.3718
Wald test of independent equations (rho = 0): chi2(1)	47.77 ***

*** and * indicate statistical significance at the 1%, and 10% levels, respectively.

**Table 4 ijerph-18-10056-t004:** Estimated coefficients: bars closures and interaction with work, income, and health.

Outcome Equation	Dependent Variable SATLIFE			
BARS_CLOSE	−2.6364 ***	−2.6231 ***	−2.3195 ***	−2.4635 ***
BARS_CLOSE * JOB_LOSS	−0.2349			
BARS_CLOSE * FUTURE_LESSINCOME		−0.1274		
BARS_CLOSE * LONELY			−0.1340	
BARS_CLOSE * HEALTH_STATUS				−0.0815
Constant	9.2474 ***	−1.3591 ***	−1.3571 ***	9.2222 ***

*** and * indicate statistical significance at the 1%, and 10% levels, respectively. All these estimations include the same controls as those reported in Table 3. Their estimated coefficients are not reported here for space constraints.

## Data Availability

The daily updated data of the LWCV survey are available for non-commercial research purposes from the data archive of the IZA—Institute of Labor Economics, see: https://datasets.iza.org/dataset/1388/living-and-working-in-coronavirus-times-survey (accessed on 5 January 2021). Institutions interested in specific countries, target groups, or in developing specific modules related to COVID-19, please contact k.g.tijdens@uva.nl or office@wageindicator.org.

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
