# Peer review of "Are Spaniards Happier When the Bars Are Open? Using Life Satisfaction to Evaluate COVID-19 Non-Pharmaceutical Interventions (NPIs)"

_ijerph, 2021, doi:10.3390/ijerph181910056_

Round 1
Reviewer 1 Report
This paper focuses on exploring the impact of bars and restaurants closure on life satisfaction in Spanish regions. The authors adopted endogenous binary-treatment regression to analyze the relationships. The analysis shows that the closure of bars and restaurants has a significant negative impact on life satisfaction.
The conclusion of this paper looks interesting to me. The analysis method is carefully designed. I am particularly impressed by the discussions that examine the impact of the interaction bar closure with work, income, and health. However, I have some concerns about this paper, as listed below:
1. The conclusion that the closure of bars and restaurants has a significant negative impact on life satisfaction is interesting. However, I am sure whether the conclusion is important and impactful. Note that the purpose of closing bars and restaurants is to slow down the spread of COVID-19. Therefore, the conclusion of this paper does not seem sufficient to convince the government to open bars and restaurants if there are a large number of daily cases. The authors might want to discuss better how the policymaker can utilize the conclusion of this paper.
2. In Section 4, the authors claim that the significant association is due to feelings of freedom and comparative nature. The interpretation looks reasonable; however, it is not clear how the authors reach this conclusion. Is there data that can support such interpretation? Is there previous research that shows that such interpretation explains similar problems?
3. Although the authors designed the study carefully, it still looks possible that some unobserved confounders exist, which might lead to potential bias in the analysis. Note that the closure of bars and restaurants happen in specific regions. Therefore, the differences in life satisfaction might be due to the different regional cultures and environments rather than the closure of bars. In addition, the local governments that keep bars open might tend to support individual freedoms. Such governments might implement several policies that support individual freedoms. Therefore, life satisfaction might be due to other policies rather than open the bars. The authors might need to discuss the limitations and potential bias of their analysis in the paper.
Minors:
4. The authors might need to provide more details of the respondents, including the percentage of the respondents in each region and the percentage of respondents who are impacted by bar closure.
5. The authors do not clarify the parameters that are involved in the maximum likelihood estimation. The parameters sigma and rho are also learned from the estimation in addition to beta and delta. Is it correct?
6. In Table 2, the coefficients for Age^2 and Cases are small. However, they are statistically significant. Is this possible? Note that the null hypothesis for the hypothesis test is the coefficient equals 0.
7. I can understand Table 3, where the authors consider one interaction term at a time and train four independent models. However, it might look confusing to some readers because all the columns are SATLIFE. Some additional explanation might be necessary for this table.
8. In most parts of this paper, the authors discuss "bars and restaurants closure." However, in the last line of Table 1, BAR_CLOSE is a variable that indicates whether bars are closed. Does BAR_CLOSE=1 also imply that restaurants are also closed?
9. There is a typo in line 158. "Minimum" should be "maximum."
Author Response
Response to Reviewer 1 Comments
Manuscrip ID: ijerph-1359582
Title: Are Spaniards happier when the bars are open? Using life satisfaction to evaluate COVID-19 non-pharmaceutical interventions (NPIs)
Thank you very much for the evaluation and your comments. Please find our responses below. We use black for your comments and red color for my responses so to facilitate reading. We use red italics when we copy specific parts of the new manuscript for the clarity of our explanations. We are grateful for the valuable comments and advice we have received that have helped us to elaborate a more consistent manuscript. We are confident the paper has considerably improved and hope it is now ready for publication. Nonetheless, if you consider it appropriate and if there are further comments from you, we are ready to take into account further suggestions.
Faithfully
Comments/responses and Suggestions/changes for/from Authors
General comment: This paper focuses on exploring the impact of bars and restaurants closure on life satisfaction in Spanish regions. The authors adopted endogenous binary-treatment regression to analyze the relationships. The analysis shows that the closure of bars and restaurants has a significant negative impact on life satisfaction.
The conclusion of this paper looks interesting to me. The analysis method is carefully designed. I am particularly impressed by the discussions that examine the impact of the interaction bar closure with work, income, and health. However, I have some concerns about this paper, as listed below.
Response to general comment: We would like to thank reviewer 1 for the general positive overview.
Point 1. The conclusion that the closure of bars and restaurants has a significant negative impact on life satisfaction is interesting. However, I am sure whether the conclusion is important and impactful. Note that the purpose of closing bars and restaurants is to slow down the spread of COVID-19. Therefore, the conclusion of this paper does not seem sufficient to convince the government to open bars and restaurants if there are a large number of daily cases. The authors might want to discuss better how the policymaker can utilize the conclusion of this paper.
Response 1: Thank you very much for this comment.
The goal of the paper is to evaluate the effect of NPIs on happiness. We use Spain and bars and restaurants closures as a case study. Although the main goal of NPIs is to slow down the spread of COVID-19, side effects occur on the economy and mental health. We have adapted the introduction and conclusions to motivate our goal and to discuss our conclusion showing how policy makers can utilize the conclusions.
In the Introduction:
- We have changed the paragraph where we explain the discussion about the effectiveness of NPIs in terms of slowing down the spread and death rate of the virus. We want to emphasise the lack of scientific consensus on whether stricter measures are more efficient in fulfilling that goal. This brings research interest to NPIs’ side effects. See this paragraph in the introduction where we set and motivate the main goal of the paper versus the primary goal of NPIs:
Second, while the first goal of the restriction measures has been to contain the spread of the virus, their socio-economic side-effects (those on life satisfaction included) are also important. There is an open discussion on the most effective policies to contain the spread of the virus. Specifically, there is a debate on whether stricter lockdown policies are associated with lower mortality [14] (Bjørnskov, 2021). Some studies report that stricter measures, such as lockdowns, are associated with lower transmissions and fewer deaths (Flaxman et al., 2020, Conyon et al., 2020) [15, 16]. Other studies report quantification problems and that less disruptive and costly NPIs can be as effective as more intrusive and drastic ones (Brodeur et al., 2021, Soltesz et al., 2020, Born, at al 2021, Atkeson et al., 2020, De Larochelambert et al., 2020, Haug et al., 2020, Brauner et al., 2021) [17, 18, 19, 20, 21, 22]. We do not aim to make a detailed literature review of NPIs´ effectiveness on the spread of the disease nor to engage in the above discussion. However, given the lack of scientific consensus on NPIs efficiency on health outcomes, it might be useful to explore their social and economic consequences and side effects. Under the uncertainty that two-policy measures are equally efficient against the spread of the virus and mortality, policy makers may choose the one with a smaller impact on life dissatisfaction. In this sense, our focus on well-being (i.e. life satisfaction) complements the evaluation of NPI’s in terms of health outcomes.
- We have included new references to better illustrate the lack of consensus on the effect of NPIs in reducing mortality:
- Flaxman, S., Mishra, S., Gandy, A. et al.Estimating the effects of non-pharmaceutical interventions on COVID-19 in Europe. Nature 584, 257–261 (2020). https://doi.org/10.1038/s41586-020-2405-7
- Conyon, Martin J. and He, Lerong and Thomsen, Steen, Lockdowns and COVID-19 Deaths in Scandinavia (June 1, 2020). Available at SSRN: https://ssrn.com/abstract=3616969or http://dx.doi.org/10.2139/ssrn.3616969
- Brodeur, A, Gray, D, Islam, A, Bhuiyan, S. A literature review of the economics of COVID-19. Journal of Economic Surveys. 2021; 35: 1007– 1044. https://doi.org/10.1111/joes.12423
- Soltesz, K., Gustafsson, F., Timpka, T. et al.The effect of interventions on COVID-19. Nature 588, E26–E28 (2020) https://doi.org/10.1038/s41586-020-3025-y
- We also explain further in the introduction that, although the mechanisms by which mental health interacts with the immune system are not yet clear, if happiness is protective, life satisfaction may be an important piece of the puzzle to evaluate NPIs and therefore may contribute to policymaking.
- The new introduction helps us to further elaborate our conclusions following reviewer’s suggestion. The usefulness for policy makers is that under the uncertainty that two-policy measures are equally efficient against the spread of the virus and mortality, policy makers may choose the one with a smaller impact on life dissatisfaction. We have reworded the first paragraph of the conclusions:
The COVID-19 NPIs aim to maximize health outcomes while minimizing their economic, social and mental health consequences. Whether stricter lockdown policies are associated with lower transmission and mortality is an open discussion. Uncertainity on whether stricter policies are more efficient in reducing mortality makes side effects in the economy and mental health relevant for policymaking. In addition, if happiness is protective, interventions targeting to minimize the negative impact on mental health and happiness may have a favorable impact on health and the evolution of the pandemic. When there is uncertainty about the relative efficiency of two policies, policy makers may choose the one with a smaller impact on life dissatisfaction. Assuming that the primary goal is to contain the virus, life satisfaction is a legitimate goal for governments and evaluating NPIs in terms of their impact on happiness is useful.
- We have also made other small changes in the discussion section aiming to focus the attention on the goal of the paper and to illustrate better how the policymaker can utilize the conclusion of this paper.
Point 2. In Section 4, the authors claim that the significant association is due to feelings of freedom and comparative nature. The interpretation looks reasonable; however, it is not clear how the authors reach this conclusion. Is there data that can support such interpretation? Is there previous research that shows that such interpretation explains similar problems?
Response 2:
Thank you for this comment. The new version illustrate better that interpretation of the positive effect is based on previous literature. There are three factors: socialization, feelings of freedom and the comparative nature of happiness.
We get to the interpretation of the positive effect of freedom from previous literature. We use two well-known references:
- Brulé, G., & Veenhoven, R. (2014). Freedom and happiness in nations: why the Finns are happier than the French. Psychology of Well-Being, 4(1), 17. https://doi.org/10.1186/s13612-014-0017-4
- Clark, A. E., & Senik, C. (2010). Who compares to whom? The anatomy of income comparisons in Europe. The Economic Journal, 120(544), 573–594. http://www.jstor.org/stable/27765788
In the previous version of the discussion section, we only refer to those references between brackets:
We believe that the positive effect of freedom on life satisfaction explains both results [42]. Those with poorer health status could decide to say at home, keeping bars open was not a source of anxiety for them. Those feeling heathier or lonely could socialize although probably not as much as they would like. Both types of people could decide taking into account their personal characteristics and circumstances. Feelings of freedom, i.e the possibility to take their own decisions, make citizens happier than their closed down counterparts.
The comparative nature of life satisfaction also offers an explanation [43]. People living in regions implementing stricter measures could witness the freedom of others. They also could see how their resignation from social life did not have a clear compensation in terms of number of cases and deaths.
In the new version of the discussion section, we explain these issues in more detail. The effect of freedom on happiness and happiness’ comparative nature are possible explanations of our results:
We believe that our results related to the well-known positive effect of individual’s feelings of freedom on life satisfaction. We base this interpretation on previous literature that has shown that all kinds of freedom are related to happiness. Freedom and feelings of freedom are able to explain differences in life satisfaction across countries [46 Brulé, G., & Veenhoven, R. (2014)]. The possibility to take their own decisions, make citizens happier. In regions where bars and restaurants were open, those with poorer health status could decide to stay at home, keeping bars open was not a source of dissatisfaction for them. Those feeling heathier or lonely could socialize although probably not as much as they would like. Both types of people could decide taking into account their personal characteristics and circumstances, which probably made them happier than their closed down counterparts.
The comparative nature of life satisfaction also offers an explanation. Literature has shown that, with respect to income, happiness is comparative. It depends on the gap between individual’s income and a benchmark to whom he/she compares with (Clark, A. E., & Senik, C. (2010) [47]. People living in regions implementing stricter measures could witness the freedom of others. They also could see how their resignation from social life did not have a clear compensation in terms of number of cases and deaths. Therefore, happiness may be also comparative with respect to freedom.
We have also included a new reference about the positive effect of socialization:
- Schmiedeberg, C., & Schröder, J. (2017) Leisure Activities and Life Satisfaction: an Analysis with German Panel Data. Applied Research Quality Life,12, 137–151. https://doi.org/10.1007/s11482-016-9458-7
And this sentence when we refer to the general positive effect:
Literature has shown that meeting with friends during leisure time has a positive effect on life satisfaction while internet and tv consumption are negatively correlated with life satisfaction [49]. Keeping bars and restaurants open increases the possibilities to meet friends and socialize which explains our results.
Point 3. Although the authors designed the study carefully, it still looks possible that some unobserved confounders exist, which might lead to potential bias in the analysis. Note that the closure of bars and restaurants happen in specific regions. Therefore, the differences in life satisfaction might be due to the different regional cultures and environments rather than the closure of bars. In addition, the local governments that keep bars open might tend to support individual freedoms. Such governments might implement several policies that support individual freedoms. Therefore, life satisfaction might be due to other policies rather than open the bars. The authors might need to discuss the limitations and potential bias of their analysis in the paper.
Response 3: The reviewer is right when indicating that some unobserved confounders may still exist. Accordingly, the section Discussion includes now the following paragraph in which we discuss limitations and related biases in our analysis.
Some limitations should be noted. Though our study has been carefully designed and the variables and the empirical strategy have been carefully chosen, some unobserved confounders might still exist that influence our results. In this sense, it is important to take into account that bars and restaurants were closed in specific regions. Though the culture towards bars and restaurants is fairly similar across the Spanish territory, there might still be some different regional cultural, political and environmental differences in life satisfaction rather than the closure of bars. In what refers to NPIs, the closure of bars and restaurants was a measure added to a series of all other types of restrictions previously implemented (mobility restrictions, capacity limits in public spaces…) and still active when bars and restaurants were closed. In addition, the regional governments that kept bars and restaurants open might tend to support individual freedoms through different policies. Therefore, it is possible that differences in life satisfaction might be due to differences in the complex regulatory environments across regions, rather than the exclusive effect of keeping bars and restaurants open/closed. Besides, we have focused on some specific time period, the second wave of the pandemic. Over the subsequent waves the importance of restrictions on life satisfaction might have changed.
Minors:
Point 4. The authors might need to provide more details of the respondents, including the percentage of the respondents in each region and the percentage of respondents who are impacted by bar closure.
Response 4: As suggested by the reviewer, in this new version of the manuscript we include a new table (in the data section, Section 2, after the description of the sample summary statistics) with the percentage of respondents in each region (new Table 1). After the inclusion of this table, the old tables have been renumbered.
Table 1 . Distribution of respondents by region
Regions |
Number of respondents |
Percent % |
Andalucia |
670 |
16.71 |
Aragón |
149 |
3.72 |
Asturias |
117 |
2.92 |
Baleares |
171 |
4.26 |
Canarias |
246 |
6.13 |
Cantabria |
67 |
1.67 |
Castilla La Mancha |
144 |
3.59 |
Castilla-León |
249 |
6.21 |
Catalonia |
589 |
14.69 |
Ceuta-Melilla |
11 |
0.27 |
Comunidad Valenciana |
351 |
8.75 |
Extremadura |
120 |
2.99 |
Galicia |
154 |
3.84 |
La Rioja |
40 |
1 |
Madrid |
608 |
15.16 |
Murcia |
134 |
3.34 |
Navarra |
76 |
1.9 |
Vasque Country |
114 |
2.84 |
4,010 |
100 |
Regarding the percentage of respondents who are impacted by bar and restaurants closure, this information was already included in the original manuscript in the variables section, when describing the variable BARS_CLOSE in lines 237-238 of the new version submitted with track changes:
12% of respondents completed the questionnaire in a moment when bars and restaurants were closed in their region while open in others.
In this new version we now also include this information in the data section to make it clearer.
Over the period of analysis, 12% of the respondents were living in a region where bars and restaurants were closed at the moment of the interview due to the restrictions imposed by the regional government.
Point 5. The authors do not clarify the parameters that are involved in the maximum likelihood estimation. The parameters sigma and rho are also learned from the estimation in addition to beta and delta. Is it correct?
Response 5: The reviewer is right. This same point was also raised by the other reviewer. We did not include this information in the initial version of the manuscript in order to make old Table 2 simple. In this new version, Table 2 (new Table 3) includes the results for the ancillary parameters ρ, σ and λ along with the likelihood-ratio test for the null hypothesis of no correlation between the treatment-assignment errors and the outcome errors:
Table 3. Endogenous treatment-regression model of life satisfaction on bars and restaurants’ closures. Estimated coefficients.
Outcome equation |
SATLIFE |
FEMALE |
0.0864 |
AGE |
-0.0335 |
AGE^2 |
0.0005* |
TERTIARY_EDUCATION |
0.2679*** |
HEALTH_STATUS |
-0.7286*** |
L_PARTNER |
0.4122*** |
L_CHILDREN |
0.0456 |
CONFINED |
0.0270 |
EXERCISE |
0.1769*** |
LONELY |
-0.5004*** |
JOB |
0.3723*** |
JOB_LOSS |
-0.8881*** |
FUTURE_LOWERINCOME |
-0.5400*** |
COVID19 |
0.2884 |
COVID19_FAMILY/FRIENDS |
0.0660 |
BARS_CLOSE |
-2.6448*** |
Constant |
9.2533*** |
Treatment equation |
BARS_CLOSE |
CASES |
0.0002*** |
Constant |
-1.3598*** |
/athrho |
0.7952 |
/lnsigma |
0.7296 |
rho |
0.6614 |
sigma |
2.0742 |
lambda |
1.3718 |
Wald test of indep. eqns. (rho = 0): chi2(1) |
47.77*** |
We also refer to the parameters in the main text: The last rows of the table report the ancillary parameters of the estimation and the likelihood-ratio test for the null hypothesis of no correlation between the treatment-assignment errors and the outcome errors. The p-value of this test clearly suggests rejecting this null hypothesis, a result which supports the chosen empirical strategy.
Point 6. In Table 2, the coefficients for Age^2 and Cases are small. However, they are statistically significant. Is this possible? Note that the null hypothesis for the hypothesis test is the coefficient equals 0.
Response 6: The reviewer is right. Thank you. We have corrected the text accordingly. The new version of the paragraph is:
And fifth, our results are only partially consistent with the U-shaped relationship between age and life satisfaction [52]. The estimated coefficients for age and its square term, are negative and positive as corresponds to a U-shaped relationship. However, in our calculations they are not significant. This is quite an interesting finding because the elderly population is more vulnerable to the pandemic.
Point 7. I can understand Table 3, where the authors consider one interaction term at a time and train four independent models. However, it might look confusing to some readers because all the columns are SATLIFE. Some additional explanation might be necessary for this table.
Response 7: We have changed the heading of Table 3 (new Table 4) to make it clearer for the reader that the dependent variable is the same: SATLIFE.
We have included a footnote underneath the table where we explain that: All these estimations include the same controls as those in Table 3. Their estimated coefficients are not reported here for space constraints.
Point 8. In most parts of this paper, the authors discuss "bars and restaurants closure." However, in the last line of Table 1, BAR_CLOSE is a variable that indicates whether bars are closed. Does BAR_CLOSE=1 also imply that restaurants are also closed?
Response 8: The reviewer is right. Thank you. We have corrected Table 1. Yes BAR_CLOSE=1 implies that restaurants are also closed.
Point 9. There is a typo in line 158. "Minimum" should be "maximum."
Response 9: The reviewer is right. Thank you. We have corrected it. In the new version, it is line 218 (new version with tracked changes).
Reviewer also point out that English language and stile are fine but required minor spell check. We have corrected them in track changes.

Reviewer 2 Report
The authors have achieved their stated objective and have clearly indicated that the findings contribute to better understanding the effects on the public happiness in Spain of closing bars and restaurants under the COVID-19 pandemic. The authors’ conclusions based on these findings are highly relevant for policymakers and other stakeholders and of great interest for researchers.
However, the following points might be corrected or checked carefully:
1) Table 2 should include the results of the likelihood-ratio test for the null hypothesis of no correlation between the treatment-assignment errors and the outcome errors (or the results for the ancillary parameters ρ and σ).
2) Although the authors pointed out that “The estimated coefficients for age and its squared term are negative and positive” (p. 8), the coefficient for age is not significant at the 10% level. In addition, although the coefficient for age squared is significant at 10%, 5% is the much more widely used criterion when the sample size is sufficiently large. Therefore, it seems to me that both coefficients should be considered insignificant and that age does not correlate with individuals’ life satisfaction.
3) Considering the results shown in Table 1 and the Appendix Table together, the two independent variables, COVID-19 and CASES, could correlate with each other.
Author Response
Response to Reviewer 2 Comments
Manuscrip ID: ijerph-1359582
Title: Are Spaniards happier when the bars are open? Using life satisfaction to evaluate COVID-19 non-pharmaceutical interventions (NPIs)
Thank you very much for the evaluation and your comments. Please find our responses below. We use black for your comments and red color for our responses so to facilitate reading. We use red italics when we copy specific parts of the new manuscript for the clarity of our explanations. We are grateful for the valuable comments and advice we have received that have helped us to elaborate a more consistent manuscript. I am confident the paper has considerably improved and hope it is now ready for publication. Nonetheless, if you consider it appropriate and if there are further comments from you, we are ready to take into account further suggestions.
Faithfully
Comments/responses and Suggestions/changes for/from Authors
General Comment: The authors have achieved their stated objective and have clearly indicated that the findings contribute to better understanding the effects on the public happiness in Spain of closing bars and restaurants under the COVID-19 pandemic. The authors’ conclusions based on these findings are highly relevant for policymakers and other stakeholders and of great interest for researchers.
Response to general comment: We would like to thank you reviewer 2 for the general positive overview.
However, the following points might be corrected or checked carefully:
Point 1: Table 2 should include the results of the likelihood-ratio test for the null hypothesis of no correlation between the treatment-assignment errors and the outcome errors (or the results for the ancillary parameters ρ and σ).
Response 1: The reviewer is right. This same point was also raised by the other reviewer. We did not include this information in the initial version of the manuscript in order to make Table 2 simple. In this new version, Table 2 includes the results for the ancillary parameters ρ, σ and λ along with the likelihood-ratio test for the null hypothesis of no correlation between the treatment-assignment errors and the outcome errors.
Table 2. Endogenous treatment-regression model of life satisfaction on bars and restaurants’ closures. Estimated coefficients.
Outcome equation |
SATLIFE |
FEMALE |
0.0864 |
AGE |
-0.0335 |
AGE^2 |
0.0005* |
TERTIARY_EDUCATION |
0.2679*** |
HEALTH_STATUS |
-0.7286*** |
L_PARTNER |
0.4122*** |
L_CHILDREN |
0.0456 |
CONFINED |
0.0270 |
EXERCISE |
0.1769*** |
LONELY |
-0.5004*** |
JOB |
0.3723*** |
JOB_LOSS |
-0.8881*** |
FUTURE_LOWERINCOME |
-0.5400*** |
COVID19 |
0.2884 |
COVID19_FAMILY/FRIENDS |
0.0660 |
BARS_CLOSE |
-2.6448*** |
Constant |
9.2533*** |
Treatment equation |
BARS_CLOSE |
CASES |
0.0002*** |
Constant |
-1.3598*** |
/athrho |
0.7952 |
/lnsigma |
0.7296 |
rho |
0.6614 |
sigma |
2.0742 |
lambda |
1.3718 |
Wald test of indep. eqns. (rho = 0): chi2(1) |
47.77*** |
We also refer to the parameters in the main text: The last rows of the table report the ancillary parameters of the estimation and the likelihood-ratio test for the null hypothesis of no correlation between the treatment-assignment errors and the outcome errors. The p-value of this test clearly suggests rejecting this null hypothesis, a result which supports the chosen empirical strategy.
Point 2: Although the authors pointed out that “The estimated coefficients for age and its squared term are negative and positive” (p. 8), the coefficient for age is not significant at the 10% level. In addition, although the coefficient for age squared is significant at 10%, 5% is the much more widely used criterion when the sample size is sufficiently large. Therefore, it seems to me that both coefficients should be considered insignificant and that age does not correlate with individuals’ life satisfaction.
Response 2: The reviewer is right. Thank you. We have corrected the text accordingly. The new version of the paragraph is:
And fifth, our results are only partially consistent with the U-shaped relationship between age and life satisfaction [52]. The estimated coefficients for age and its square term, are negative and positive as corresponds to a U-shaped relationship. However, in our calculations they are not significant. This is quite an interesting finding because the elderly population is more vulnerable to the pandemic
Point 3: Considering the results shown in Table 1 and the Appendix Table together, the two independent variables, COVID-19 and CASES, could correlate with each other.
Response 3: Taking into account reviewer’ concern, we have checked whether the variables COVID19 and CASES could be correlated.
The correlation coefficient between these two variables is 0.0225, close enough zero to consider that the correlation is almost null.
In addition, we have tested whether there are any statistically significant differences in the means of the variable CASES between the two groups defined by the variable COVID. We have run both parametric (t-test) and non-parametric tests (Kruskal-Wallis equality-of-populations rank test).
In both cases the p-values are well over the threshold of the 10%: the p-value of the t-test is 0.15 and that of the Kruskal-Wallis test is 0.96. Hence, we fail to reject the null hypothesis of the equality of means between the two groups.
All these results suggest that the correlation between COVID19 and CASES does not raise any serious issue.

Round 2
Reviewer 1 Report
The authors have made a notable effort to improve the paper and have addressed all my concerns. I do not have additional comments.